# Chemical Recycling of Used Motor Oil by Catalytic Cracking with Metal-Doped Aluminum Silicate Catalysts

**Daniela Almeida Streitwieser** [1,2,*] **, Arturo Arteaga** [1] **, Alvaro Gallo-Cordova** [3] **, Alexis Hidrobo** [1] **and Sebastian Ponce** [1,*]

[1] Department of Chemical Engineering, Universidad San Francisco de Quito USFQ, Diego de Robles s/n y Av. Interoceánica, Quito 170157, Ecuador; arturo.arteaga@alumni.usfq.edu.ec (A.A.); ahidrobo@usfq.edu.ec (A.H.)

[2] Faculty Life Sciences, Reutlingen University, Alteburgstr. 150, 72762 Reutlingen, Germany

[3] Department of Nanoscience and Nanotechnology, Instituto de Ciencia de Materiales de Madrid, ICMM/CSIC, C/Sor Juana Inés de la Cruz 3, 28049 Madrid, Spain; alvaro.gallo@csic.es

[*] Correspondence: d.almeida_streitwieser@reutlingen-university.de (D.A.S.); sponce@usfq.edu.ec (S.P.)

**Abstract:** The chemical recycling of used motor oil via catalytic cracking to convert it into secondary diesel-like fuels is a sustainable and technically attractive solution for managing environmental concerns associated with traditional disposal. In this context, this study was conducted to screen basic and acidic-aluminum silicate catalysts doped with different metals, including Mg, Zn, Cu, and Ni. The catalysts were thoroughly characterized using various techniques such as $N_2$ adsorption–desorption isotherms, FT-IR spectroscopy, and TG analysis. The liquid and gaseous products were identified using GC, and their characteristics were compared with acceptable ranges from ASTM characterization methods for diesel fuel. The results showed that metal doping improved the performance of the catalysts, resulting in higher conversion rates of up to 65%, compared to thermal (15%) and aluminum silicates ($\approx$20%). Among all catalysts, basic aluminum silicates doped with Ni showed the best catalytic performance, with conversions and yields three times higher than aluminum silicate catalysts. These findings significantly contribute to developing efficient and eco-friendly processes for the chemical recycling of used motor oil. This study highlights the potential of basic aluminum silicates doped with Ni as a promising catalyst for catalytic cracking and encourages further research in this area.

**Keywords:** aluminum silicate; metal doping; used motor oil; cracking; chemical recycling

## 1. Introduction

The problem of soil, air, and water contamination caused by used lubricating oil is a major environmental concern in the modern world [1,2]. This issue has intensified over the last few decades, and finding a sustainable solution for the proper disposal of used motor oil (UMO) has become a challenge. UMO typically contains toxic substances, such as heavy metals, polycyclic aromatic hydrocarbons (PAHs), and other hazardous chemicals that can cause severe environmental and health problems [3]. The adverse effects of these substances on human health are well-documented, with evidence of carcinogenic, mutagenic, and reproductive effects [4–7].

It is estimated that, globally, about 20 million tons of UMO are produced annually, highlighting the need for sustainable waste management technologies to address this challenge. Such technologies will help protect the environment, conserve resources, provide economic benefits, and ensure compliance with legal requirements. Developing sustainable waste management technologies for UMO is crucial for achieving a more sustainable and resilient future.

Various standard waste treatment processes have been explored to address the issue of used motor oil (UMO), including incineration, combustion [8], pyrolysis, and

co-pyrolisis [9–11]. However, incineration and combustion processes suffer from high investment costs and environmental pollution, making them less efficient options for UMO treatment. Pyrolysis, on the other hand, is considered a more efficient approach, as it produces reusable products such as gases, oils, and carbonaceous residues that can be used as fuels [7,12]. Co-pyrolysis is also an interesting alternative for co-processing different unwanted wastes together with UMO. For instance, Arilla-Suarez et al. [13] co-processed waste surgical masks, UMO, and biomass together. They showed high yields of an oily product with a hydrocarbon content in the diesel range. Upon further analysis of these products, it was determined that they meet regulatory standards for quality in regard to parameters such as high heating value (HHV), viscosity, density, and sulfur content. Other successful examples in the literature are the co-pyrolysis of used motor oil with tetra pack [14] and low-density polyethylene [15] wastes.

Catalytic cracking, a more advanced pyrolysis technique, offers even more advantages for UMO treatment. This well-studied method allows the conversion of heavy residues by breaking the large hydrocarbon molecules, producing lighter components lower in sulfur, and leaving coke behind [16]. Clays, amorphous silica, and zeolites have typically been used to improve the reaction rate and efficiency of lighter products [17]. In particular, mesoporous silica has shown advantages for materials diffusion and its application in catalytic cracking, improving the selectivity to shorter hydrocarbon chains [18]. They are also attractive due to the possibility of surface modification and control of pore distribution. They can be used as support for metal impregnation into the silica structure to increase the catalytic activity [19]. In particular, metals such as Zn, Ni, and Cu have been widely used to modify porous matrices for changing textural properties and acidity of pristine materials. Metal incorporation can implement defect structures, such as lattice distortion and abundant oxygen vacancies, that might be beneficial for boosting the catalyst's activity [20]. For instance, recently, zinc-loaded ZSM-5 catalysts have shown to be highly efficient for the catalytic cracking of camelina oil into hydrocarbon fuel [21] and biodiesel into green aromatics [22]. Due to their high capacity for carbon adsorption, nickel-based catalysts are also highly active catalysts for hydrocarbon cracking [23,24].

For UMO, catalytic cracking converts the residue into secondary diesel-like fuels (DLFs) using significantly lower temperatures than traditional thermal pyrolysis methods [25]. DLFs have similar physicochemical and rheological properties as commercial diesel, which makes them an attractive option for use in diesel engines without modification. For instance, Maceiras et al. [26] studied the use of the two additives sodium hydroxide and sodium carbonate to purify the obtained fuel to reduce its sulfur content. Moreover, quantitative gas chromatography showed that obtained diesel contained 63 % of total hydrocarbons and small amounts of benzene.

In addition to their compatibility with diesel engines, DLFs have other benefits, such as avoiding flow and ignition problems. These fuels can be used as a substitute for commercial diesel, reducing the need for fossil fuels, and promoting sustainable energy solutions. Therefore, the use of catalytic cracking for UMO treatment holds great promise for promoting a circular economy and addressing the environmental and health concerns associated with UMO disposal.

A previous study had investigated the effectiveness of mesoporous aluminum silicates (Al/Si) as catalysts for the catalytic cracking of waste motor oil [25]. The study had revealed that the catalytic processes showed a pseudo-first-order rate equation concerning the concentration of UMO, indicating that the rate of the reaction depended only on the concentration of UMO.

Furthermore, compared to thermal processes, catalytic processes showed lower activation energies and increased overall yield to liquid fuel from 63% to 90%, demonstrating the efficiency of the catalytic cracking method. The study had also presented the first insights into the benefits of doping the Al/Si matrix with metals like zinc.

Interestingly, the catalytic cracking process produced a diesel-like fuel that met the requirements for Diesel II. This finding is significant, as it demonstrates that the catalytic

cracking of UMO has the potential to produce high-quality fuels that can be used in existing diesel engines without modifications. Overall, these findings provide valuable insights into the potential of catalytic cracking as a sustainable and efficient approach for UMO treatment and fuel production.

In this new work, the scope of the investigation has been extended to include a more comprehensive study of the catalytic cracking reaction of UMO using mesoporous aluminum silicate catalysts prepared under both acidic and basic conditions and doped with different metals, including Mg, Zn, Cu, and Ni. To ensure a thorough comparison of the various processes, complete raw materials and product characterization have been developed. This will enable a detailed analysis of the differences between thermal, catalytic, and metal-doped catalytic processes in terms of the products obtained and their properties.

The use of different metal dopants will also provide insight into the effects of various metals on the catalytic performance of the Al/Si matrix. By studying the performance of these catalysts under both acidic and basic conditions, it will be possible to compare the effectiveness of each condition on the catalytic cracking of UMO.

## 2. Materials and Methods

### 2.1. Raw Materials

Used motor oil (UMO) was collected from a local mechanical workshop in Quito, Ecuador. Collected UMO was pre-treated by filtering the oil with d80 and d60 metal meshes for the removal of solid particles. Then, the moisture of the UMO was removed via heating at 100 °C for 1 h under continuous agitation (80 rpm).

### 2.2. Cracking Process

Pre-treated UMO was thermally cracked in a Precision Scientific Petroleum Herzog distillation unit (see Figure 1). Then, 100 mL of pre-treated UMO was filled in a 250 mL borosilicate glass flask with boiling bulbs. First, the solution was pre-heated for 5 min up to a temperature of 100–110 °C; then, the final temperature was set between 370 and 415 °C and kept for 180 min. Non-reacted raw material remained in the flask, while reaction gases exited the reactor to be either condensed or collected as non-condensable gases. The reaction gases were cooled to 30 °C and bubbled through a liquid trap. The condensed liquids were collected and quantified in the modified graduated cylinder, while samples of the gaseous products were collected in a hermetic Tedlar gas sampling bags. The liquid product was weighed during the experiment as a function of the average temperature reached. For catalytic cracking, 1 g of prepared catalysts was added to the glass container before starting the cracking process. The experimental design and abbreviations used in this work are presented in Table 1.

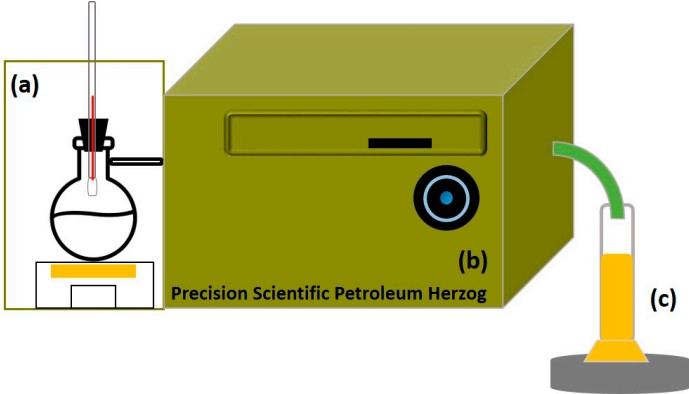

**Figure 1.** Schematic representation of the experimental setup. (**a**) Glass batch reactor, (**b**) heater section, and (**c**) graduated cylinder for the collection of the product [25].

**Table 1.** Experimental design of this study.

| Experiment | Denomination | Impregnated Metals | Percentage of Metal in Catalyst | Reaction Temperature (°C) |
|---|---|---|---|---|
| Thermal cracking | Thermal | - | - | 375–415 |
| Catalytic cracking using Al/Si under acidic and basic preparation | A-Al/Si/ B-Al/Si | - | - | 380–390 |
| Catalytic cracking using Al/Si acid preparation with metal-doped | A-Al/Si-Me | Zn-Mg-Cu-Ni | 1% | 380–390 |
| Catalytic cracking using Al/Si basic preparation with metal-doped | B-Al/Si-Me | Zn-Mg-Cu-Ni | 1% | 380–390 |

For all experiments, conversion, yield, and selectivity were calculated using the mass balance as presented in Equation (1).

$$m_o = m_G + m_L + m_R \tag{1}$$

where $m_o$ represents the initial amount of UMO, $m_G$ is the amount of gaseous products collected in the gas trap, $m_L$ is the amount of liquid products collected in the graduated cylinder, and $m_R$ is the unreacted UMO at the glass reactor. For this process, all losses were considered as gaseous products in the mass balance.

Conversion ($X$), Yield ($Y_i$), and Selectivity ($S_i$) were calculated according to Equations (2)–(4).

$$X = \left( \frac{m_O - m_R}{m_o} \right) \times 100 \ [\%] \tag{2}$$

$$Y_i = \left( \frac{m_i}{m_o} \right) \times 1100 \ [\%] \tag{3}$$

$$S_i = \left( \frac{m_i}{m_O - m_R} \right) \times 100 \ [\%] \tag{4}$$

where $i$ represents the gaseous or liquid products.

*2.3. Catalysts Synthesis*

Mesoporous aluminum silicates were synthetized according to the following procedure as presented in Vargas et al. [25]. Aluminum tri-sec-butylate (TBA) and tetraethyl orthosilicate (TEOS) were used as the aluminum precursor and the silicon–organic precursor, respectively, with a molar relation between silicon and aluminum of 7.5. Triton X-114 was used as a pore modulator agent. Iron, copper, zinc, magnesium, and nickel salts were added according to the defined relation for metal doping. In summary, 4.5 g of Triton X-114 were vigorously agitated with 30 mL of deionized water and the metallic salt (1%). For the acid and basic method, the pH of the mixture was controlled to 1 and 10, by adding hydrochloric acid or ammonia, respectively. Then, 11.5 g of TEOS were added and agitated vigorously for 2 h. Subsequently, 1.9 g of TBA were added and kept in agitation for 24 h. The resulting mixture was transferred to a hydrothermal Teflon reactor, where it was kept at 150 °C for 24 h. The solid product was then mixed with ethanol and dried at atmospheric temperature. Finally, the dry product was calcined at 550 °C for 24 h.

*2.4. Characterization Methods*

Gas chromatography was applied to characterize the gaseous and liquid distillation products. A Thermo Scientific TRACE 1310 Gas Chromatograph with a split/splitless injector and an FID detector was used. The samples were injected with a sample time of 1.00–22.00 min at an inlet temperature of 80 °C and helium as carrier gas. A TG Bond Q

Thermo Scientific 15 m $\times$ 0.53 mm $\times$ 20 $\mu$m with a nonpolar, 100% divinyl benzene phase column was used. The temperature in the oven was held at 30 °C for 5 min and then heated to 200 °C at a rate of 10 °C min$^{-1}$. The column rate was 2.1 mL min$^{-1}$ with a split flowrate of 5 mL min$^{-1}$ and split ratio of 2. The temperature in the detector was set at 200 °C. Two analytical standards for qualitative analysis were used. The first was a Scott Gas Cat. No. 22566, containing the gases methane, ethane, ethylene, acetylene, propane, propylene and n-butane with a concentration of 15 ppm each in nitrogen; the second standard was a Scott Gas Cat. No. 501662 contains acetylene, carbon dioxide, carbon monoxide, ethane, ethylene, and methane with a concentration of 15 ppm each in nitrogen.

The catalyst's morphology was analyzed using a JEOL JSM-IT300 scanning electron microscope and the program MP-96040EXCS External Control Software. Aluminosilicate samples were observed at LVSED with 50 Pa and 5 kV. Moreover, mapping of the samples was developed using IT300 (LA) at 10 kV. Distribution of the metal components was also carried out. $N_2$ adsorption–desorption isotherms were obtained using a Micrometrics TriStar 3000 model. All catalysts were de-gasified for at least 12 h at 393 K and 0.1 mbar and surface area was estimated by using the BET equation. Thermal gravimetric analysis was developed in a Q2000 differential scanning calorimeter, where the samples were heated in air from room temperature to 900 °C at 10 °C/min. IR spectra were obtained from a Bruker Vertex 70 V with a 2 cm$^{-1}$ resolution. Finally, a $CO_2$ temperature programmed desorption test ($CO_2$-TPD) was carried out using a Micromeritics Autochem II 2920 instrument to determine the basicity of the samples. Approximately 0.1 g of activated catalyst was introduced and treated at 180 °C (increasing temperature at a rate of 15 °C/min) for 1 h in a stream of helium (25 mL/min). Subsequently, the sample was saturated with 10% $CO_2$ for 2 h at 50 °C. After purging with helium for 30 min to eliminate $CO_2$ in the gas phase and physisorbate, the sample was heated from 50 °C to 600 °C with a ramp of 10 °C/min. The desorbed $CO_2$ was entrained by a helium stream (25 mL/min) as a carrier, thus obtaining a TCD signal when compared with a reference helium stream.

## 3. Results

### 3.1. Catalysts Characterization

The morphology of the synthesized catalysts was analyzed via SEM imaging (see exemplary images of Cu-doped basic and acid catalysts in Figure 2). On the one hand, the Cu-doped basic catalyst showed particle size between 300–500 nm with a sandy, porous, and foamy morphology (see Figure 2a). For acid catalysts, a different morphology compared to basic catalysts was observed. It is known that the presence of acidic conditions is more likely to result in the formation of aggregated particles compared to basic conditions. Protonation of the surface sites in an acidic medium might lead to stronger electrostatic interactions between particles [27]. Moreover, the dissolution and re-precipitation of aluminum and silicon species might be more pronounced. The re-precipitation of these ions onto the particle surfaces can contribute to the formation of aggregates by bridging particles together. These forces can promote particle aggregation and the formation of large clusters, as observed (>1 $\mu$m). This aggregation seems to produce a more rigid structure with a smooth surface, although still with the presence of tiny pores (see Figure 2b). For all materials, Al, Si, Ni, Mg, and Cu were successfully detected in metal-doped catalysts through EDS examination (see an exemplary result in Figure S1 in Supporting Information (SI)).

Further catalyst characterization is displayed in Figure 3, where $N_2$ isotherms, FTIR, and TGA analyses are included. Textural characteristics were obtained from $N_2$ adsorption–desorption isotherms (see Figure 3a).

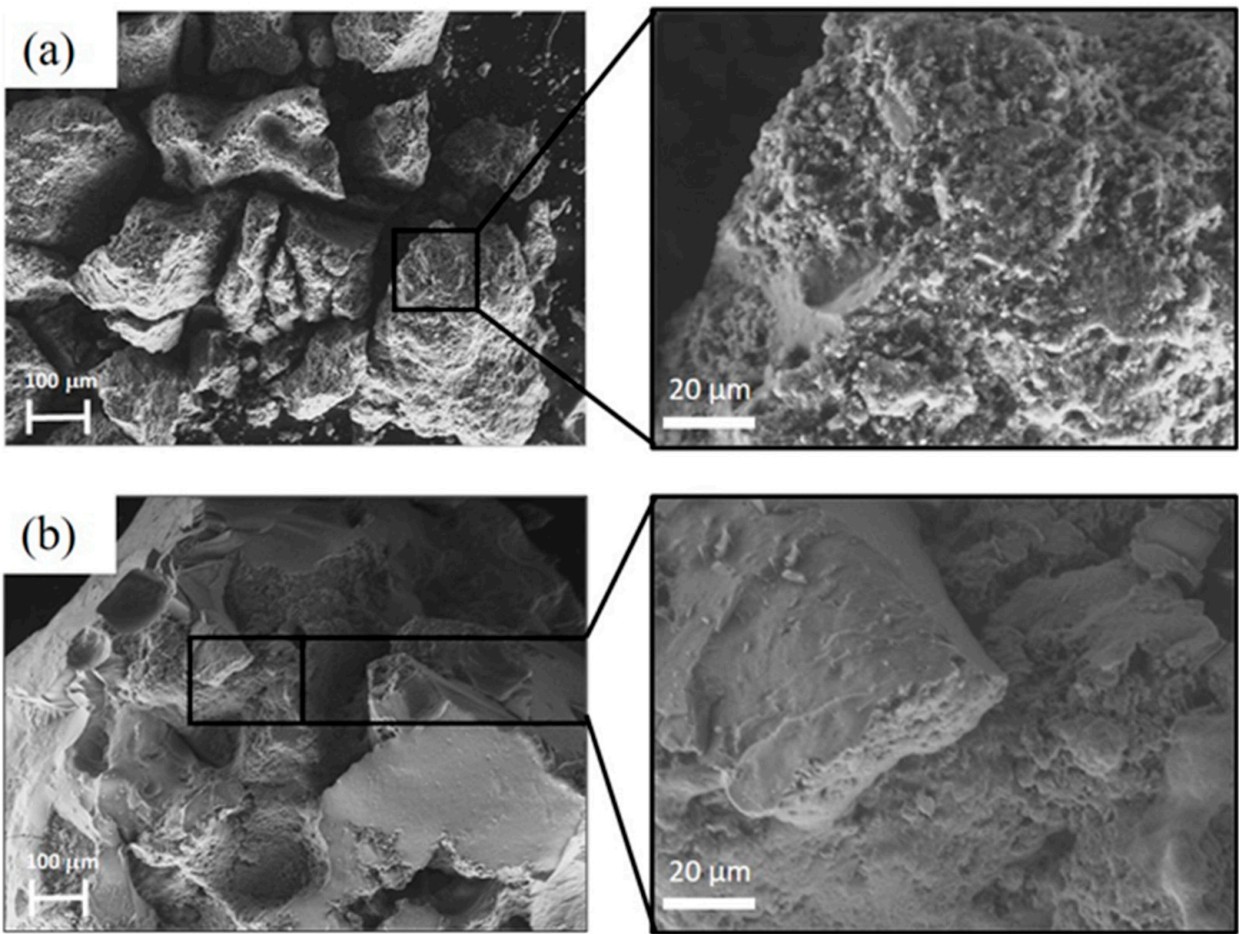

**Figure 2.** SEM images of (**a**) a basic Cu-doped aluminosilicate and (**b**) an acid Cu-doped aluminum silicate catalyst.

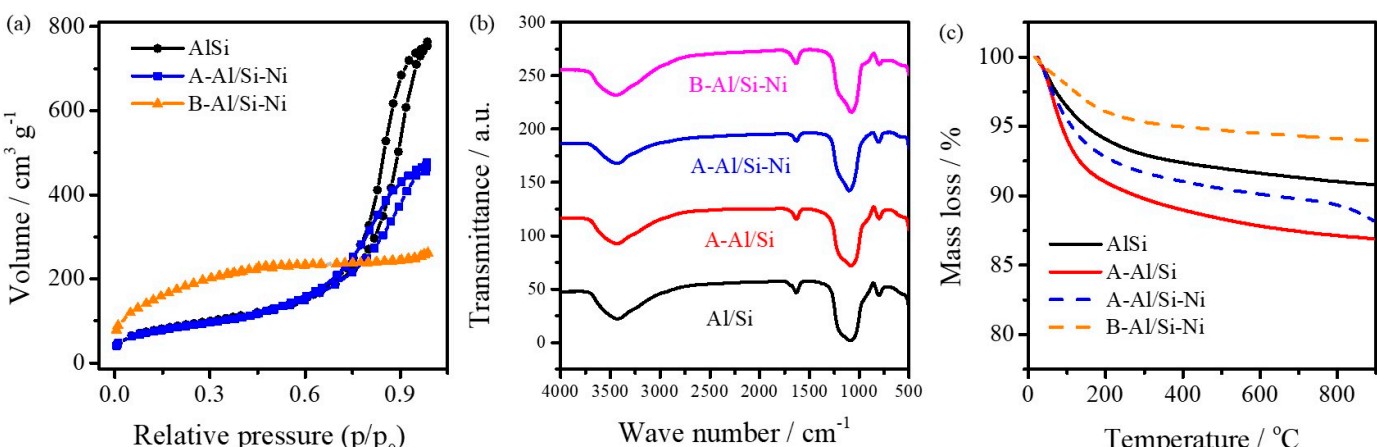

**Figure 3.** Exemplary (**a**) $N_2$ adsorption–desorption isotherms, (**b**) infrared spectra, and (**c**) thermal gravimetric analyses of synthesized materials.

All synthesized materials exhibited a mesoporous texture, i.e., pore diameters under 50 nm (See all values in Table 2). For Al/Si, B-Al/Si, A-Al/Si-X (X = Mg, Zn, Cu, Ni), B-Al/Si-Y, (Y = Zn, Cu), a type IV isotherm was observed (Please refer to Figure S2 in SI), which is typical for adsorption on mesoporous solids with a narrow pore size distribution. These materials show pore diameters around 10 nm, except for B-Al/Si-Zn, which shows a shift to 20 nm. Among them, metal-doped materials showed a reduction in their surface

area compared to Al/Si catalysts (e.g., Al/Si = 310 m$^2$ g$^{-1}$, B-Al/Si-Zn = 104 m$^2$ g$^{-1}$, see all values in Table 2). On the other hand, A-Al/Si, and B-Al/Si-Mg clearly showed different behavior, with an increment in the specific surface area and a reduction of the pore size diameter. This behavior is even more noticeable for B-Al/Si-Ni, which follows an isotherm of type I, typical for microporous solids. This behavior is related to the highest specific surface area (642 m$^2$ g$^{-1}$) and smallest pore sizes (2.5 nm) measured.

**Table 2.** BET, volume and average pore diameter values for synthesized materials.

| Sample | | BET (m$^2$ g$^{-1}$) | Volume Mesopore (cm$^3$ g$^{-1}$) | Average Pore Diameter (nm) |
|---|---|---|---|---|
| Al/Si | | 310 | 1.15 | 12.9 |
| A-Al/Si | | 490 | 0.71 | 5.81 |
| B-Al/Si | | 190 | 0.67 | 12.82 |
| | A-Al/Si-metal | | | |
| | Mg | 246 | 0.95 | 11.31 |
| | Zn | 132 | 0.42 | 13.07 |
| | Cu | 218 | 0.66 | 12.45 |
| | Ni | 298 | 0.74 | 8.98 |
| | B-Al/Si-metal | | | |
| | Mg | 394 | 0.6 | 6.12 |
| | Zn | 104 | 0.49 | 20.79 |
| | Cu | 122 | 0.36 | 13.41 |
| | Ni | 642 | 0.24 | 2.53 |

Moreover, FTIR spectra were recorded from 250 to 4000 cm$^{-1}$ for all synthesized catalysts. As depicted in Figure 3b, the nature of the catalyst seems not to be affected by metal doping, as similar spectra were obtained. The 3440 and 1480 cm$^{-1}$ bands reflect the presence of water at the material's surface, while vibrations in the zone between 500 and 1480 cm$^{-1}$ are characteristic of metal silicates, e.g., O-Si-O and Al-O-Si oscillations [28]. Finally, as expected, all synthesized catalysts showed similar behavior after thermal gravimetric analysis (see Figure 3c). A mass loss of around 10% above 200 °C is observed, which can be related to the dehydroxylation of the aluminosilicate framework [29]. Supporting information includes the N$_2$ adsorption–desorption isotherms (Figure S2), TGA (Figure S3), and FTIR spectra (Figure S4) of all catalysts.

As shown above, B-Al/Si-Ni materials showed the most promising characteristics for increasing the catalytic activity (i.e., higher surface areas and smaller pore sizes). Thus, the basicity of the Al/Si and B-Al/Si-Ni catalysts was investigated using CO$_2$-TPD for comparison. Figure 4 displays the CO$_2$ desorption profiles of both catalysts, allowing for the identification of higher basicity based on the peaks' intensities. In the case of Al/Si, a narrow peak was observed at 75 °C, whereas the B-Al/Si-Ni catalyst exhibited two sharp peaks at 89 and 221 °C, with the latter being the most pronounced. This suggests that the basicity of the B-Al/Si-Ni catalyst is stronger than that of Al/Si. Moreover, the active sites can be classified as weak, medium, strong, and very strong, based on desorption temperatures: 150–350 °C, 350–500 °C, and >500 °C, respectively. The number of weak basic sites for Al/Si was determined to be 0.15 mmol/g, which is comparable to B-Al/Si-Ni (0.21 mmol/g). However, the B-Al/Si-Ni catalyst exhibited an additional 0.35 mmol/g of medium basic sites, demonstrating its higher basicity.

### 3.2. Characterization of Raw Material and Cracking Products

Raw materials and products have been characterized to identify their potential use as commercial diesel-like fuel. Table 3 presents the acceptable ranges for Diesel II, a liquid fuel with a reduced sulfur content used for combustion engines, according to the Ecuadorian technical norm NTE INEN 1489. Results of the analysis of the UMO and ranges for cracking products are presented in Table 3 point parameters.

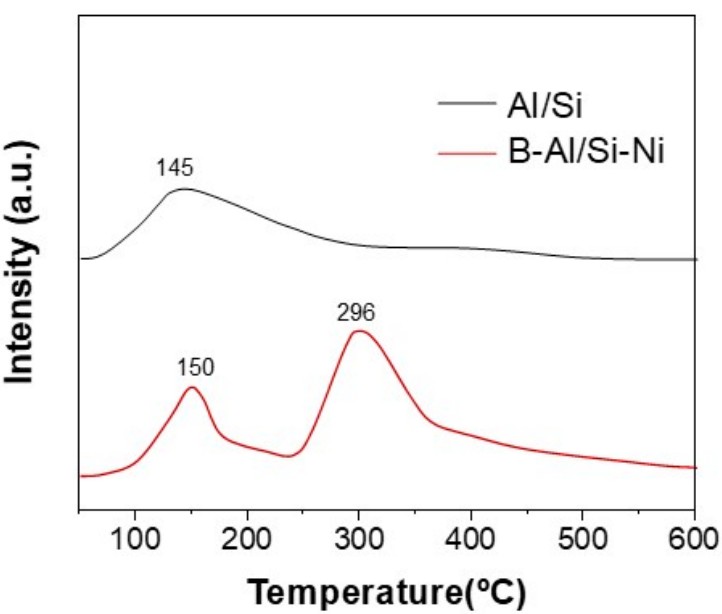

**Figure 4.** $CO_2$-TPD curves of Al/Si and B-Al/Si-Ni catalysts.

**Table 3.** Characterization of UMO and Products from the cracking process.

| Property | Method | INEN 1489:2012 Diesel II | Raw Material UMO | Ther-mal | A-Al/Si | B-Al/Si | A-Al/Si-Co | B-Al/Si-Ni |
|---|---|---|---|---|---|---|---|---|
| Density at 15 °C (g ml⁻¹) | ASTM D1298 | ~0.87 | 0.88 | 0.84 | 0.81 | 0.82 | 0.82 | 0.82 |
| Kin. Viscosity at 40 °C (cSt) | ASTM D2270 | 2.5–5.0 | 101.4 | 3.2 | 1.6 | 2.0 | 1.9 | 2.4 |
| API Degree at 15 °C | ASTM 1298 | ~33.00 | 28.82 | 37.12 | 43.09 | 41.07 | 41.86 | 41.09 |
| 90% Distillation Temp. (°C) | ASTM D86 | Less than 360 | - | 280 | 320 | 310 | 315 | 318 |
| Ignition Point (°C) | ASTM D56 | 51 | - | 66 | 51 | 46 | 41 | 51 |

The results obtained from the analysis clearly show that the UMO samples used in the study do not meet the permissible values for a Diesel II. This is not surprising, given that UMO typically contains harmful substances such as heavy metals and polycyclic aromatic hydrocarbons (PAHs) that can negatively impact fuel quality. Table 3 shows the product values obtained after both thermal and catalytic cracking. While the values are closer to those expected for Diesel II, further processing may be required to comply with norm requirements fully. This is mainly due to higher amount of volatile compounds observed in the API degree, which can affect the fuel's stability and ignition properties. Other rectification processes could be applied to remove these compounds further and improve fuel quality. Alternatively, the use of metal-doped Al/Si catalysts could potentially reduce the presence of such compounds during the catalytic cracking process, resulting in a product that more closely meets Diesel II specifications.

Gas analysis is an essential tool for understanding the nature of gaseous products during the catalytic cracking of waste motor oil. In this study, gas chromatography with a flame ionization detector (GC-FID) was employed to determine the composition of the gaseous products. The chromatograms of the commercial standards were used as a reference to identify the peaks in the experimental results. As shown in Figure 5a, eight peaks were identified, corresponding to methane, ethane, acetylene, ethylene, propane, propylene, n-butane, and an unknown peak (number 5). Interestingly, the temperature and type of cracking did not significantly impact the composition of the gaseous products, indicating the stability of the catalytic cracking process. The identification of these gaseous products is crucial for the optimization of the process and the development of sustainable waste management technologies.

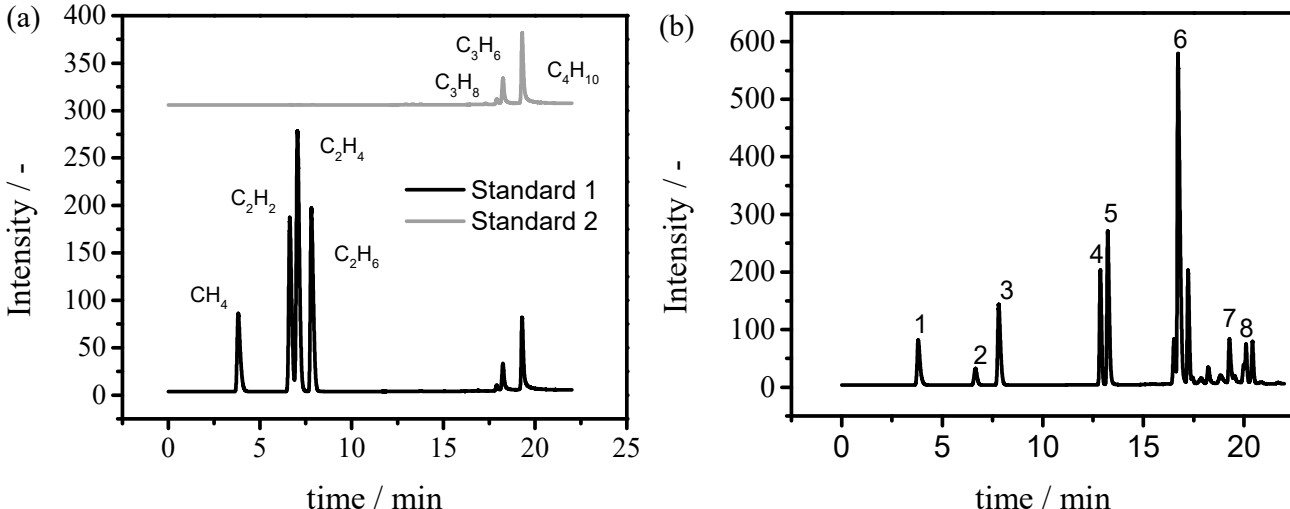

**Figure 5.** Chromatograms by GC−FID of (**a**) two analytical standards and (**b**) an exemplary catalytic cracking experiment at 390 °C with the peaks: 1. Methane, 2. Ethane, 3. Acetylene, 4. Ethylene, 6. Propane, 7. Propylene, and 8. N-butane.

### 3.3. Cracking Process

Before catalytic cracking was developed, the ideal reactor conditions for obtaining the best combination of yield and liquid products were tested. Different reactor temperatures between 370 and 415 °C were applied. The reactor temperature and recovered liquid volume were recorded as a function of time, as shown in Figure 6 for experiments at ≈370 and 390 °C.

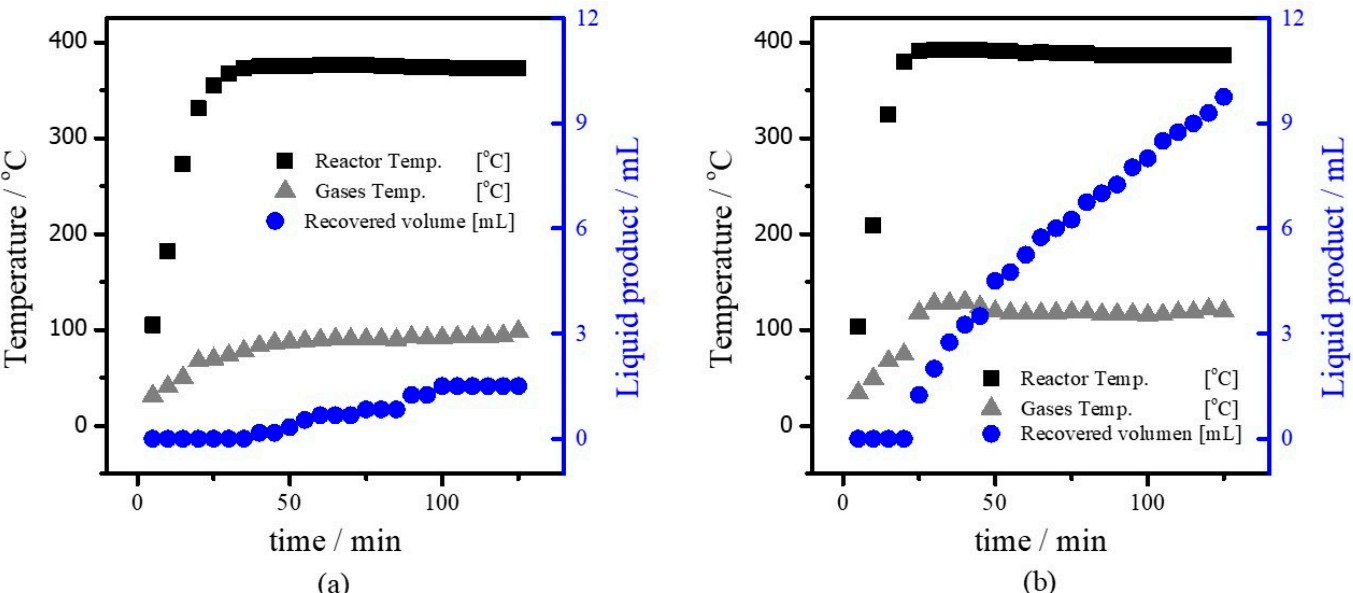

**Figure 6.** Thermal cracking performance at reactor temperature (**a**) 370 and (**b**) 390 °C.

In all experiments, after 20 min, the reactor and gas temperatures were kept constant during the cracking process. The liquid product tends to increase by incrementally increasing the reactor temperature. Nevertheless, for lower temperatures (<380 °C), liquid production was shown to be slow and intermittent (see the gray curve in Figure 6a). In comparison, the liquid production rate seems constant at higher temperatures during the cracking procedure. Nevertheless, for temperatures over 400 °C, it was observed that the whole sample volatilized and was recovered in the liquid products without being cracked.

To ensure the production of high-quality liquid products, this study performed thermal and catalytic cracking at temperatures below 400 °C. The findings are crucial in providing valuable insights into the optimal operating conditions required for maximizing the liquid product yield while avoiding undesirable volatilization of the sample.

Table 4 lists the conversion, yield, and selectivity values for thermal and acidic or basic catalytic cracking of UMO for temperatures under 400 °C. Please refer to Table S1 in SI for the mass balance results. As expected, acidic and basic catalysts increased the conversion and the liquid products yield compared to non-catalyzed thermal cracking (see Figure 7a,b). The basic catalysts are the most efficient for UMO cracking and forming liquid products with an increment of ≈10% in both conversion and liquid products yield. It is known that UMO can be enhanced in an alkali environment as it facilitates the cleavage of carbon–carbon bonds, leading to increased cracking efficiency. Moreover, basic catalysts might benefit from removing these contaminants by forming stable complexes. For all experiments, higher conversions were obtained when increasing the reaction temperature.

**Table 4.** Conversion, Yield, and Selectivity of thermal and acid–basic thermal cracking.

| Reactor Temperature (°C) | Conversion (%) | Liquid Products Yield (%) | Gaseous Products Yield (%) | Liquid Products Selectivity (%) | Gaseous Products Selectivity (%) |
|---|---|---|---|---|---|
| Thermal cracking | | | | | |
| 380 | 5.89 | 2.62 | 3.26 | 45 | 55 |
| 385 | 8.44 | 5.81 | 2.63 | 69 | 31 |
| 390 | 15.50 | 11.50 | 3.99 | 74 | 26 |
| A-Al/Si- catalytic cracking | | | | | |
| 380 | 9.28 | 4.19 | 5.10 | 45 | 55 |
| 385 | 10.60 | 6.73 | 3.87 | 63 | 37 |
| 390 | 20.58 | 13.91 | 6.66 | 68 | 32 |
| B-Al/Si- catalytic cracking | | | | | |
| 380 | 4.13 | 1.29 | 2.84 | 31 | 69 |
| 385 | 8.08 | 4.91 | 3.17 | 61 | 39 |
| 390 | 23.46 | 21.81 | 1.65 | 93 | 7 |

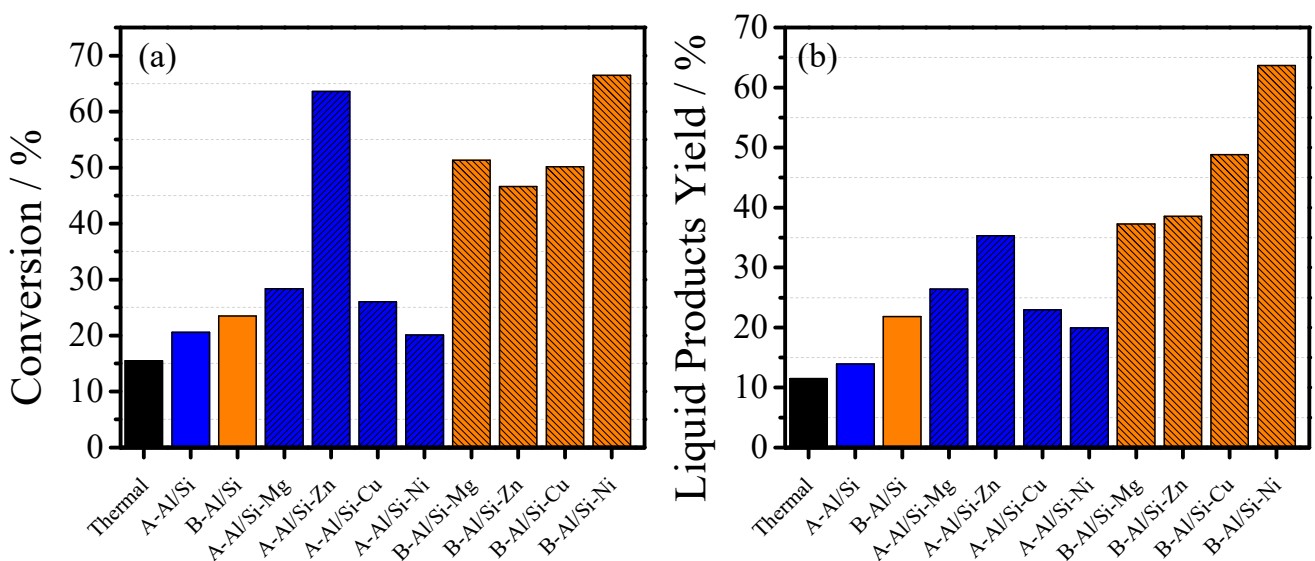

**Figure 7.** Conversion (**a**) and Liquid Products Yield (**b**) summary for all experiments.

Interestingly, selectivity towards liquid products is highly dependent on the reaction temperature and can be enhanced by the presence of a catalyst. For example, at a temperature of 390 °C, the selectivity towards liquid products is 74% and 90% for thermally and basic-catalyzed reactions, respectively. As mentioned above, basic catalysts can also favor

the production of desired lighter hydrocarbon fractions. The improved selectivity towards liquid products obtained by optimizing the reaction temperature and using a basic catalyst can have significant implications for the practical application of the method, leading to higher efficiency, better product quality, and potentially lower environmental impact.

Finally, catalytic cracking with a metal-doped catalyst was performed. Please refer to Table 5 for listed values and Table S2 in SI for mass balance results. Metal doping generally affects UMO conversion compared to simple thermal and acidic or basic catalytic cracking (see Figure 7a,b). Conversion and yield were almost three times as high as the experiments without metal doping. For instance, adding Zn in an acid Al/Si matrix shows a considerable increase in conversion and yield. While Mg, Cu, and Ni did not show a significant increment. On the other hand, all metal doping showed an increment in conversion and yield for basic Al/Si matrixes. The results indicate that metal doping can enhance the activity of the catalyst, especially in basic matrices.

**Table 5.** Conversion, Yield, and Selectivity of metal-doped acid–basic catalytic cracking.

| Reactor Temperature (°C) | Metal | Conversion (%) | Liquid Products Yield (%) | Gaseous Products Yield (%) | Liquid Products Selectivity (%) | Gaseous Products Selectivity (%) |
|---|---|---|---|---|---|---|
| | | | A-Al/Si-metal catalytic cracking | | | |
| 385 | Mg | 17.82 | 15.63 | 2.19 | 88 | 12 |
| 390 | | 28.36 | 26.39 | 1.97 | 93 | 7 |
| 385 | Zn | 17.69 | 17.67 | 0.02 | 100 | 0 |
| 390 | | 63.55 | 35.29 | 28.26 | 56 | 44 |
| 385 | Cu | 13.61 | 12.45 | 1.16 | 91 | 9 |
| 390 | | 26.00 | 22.95 | 3.05 | 88 | 12 |
| 385 | Ni | 15.15 | 12.28 | 2.87 | 81 | 19 |
| 390 | | 20.11 | 19.96 | 0.15 | 99 | 1 |
| | | | B-Al/Si-metal catalytic cracking | | | |
| 385 | Mg | 12.19 | 11.23 | 0.96 | 92 | 8 |
| 390 | | 51.30 | 37.28 | 14.02 | 73 | 27 |
| 385 | Zn | 21.81 | 20.77 | 1.04 | 95 | 5 |
| 390 | | 46.58 | 38.57 | 8.01 | 83 | 17 |
| 385 | Cu | 16.05 | 12.89 | 3.16 | 80 | 20 |
| 390 | | 50.12 | 48.83 | 1.29 | 97 | 3 |
| 385 | Ni | 20.87 | 17.45 | 3.42 | 84 | 16 |
| 390 | | 66.49 | 63.66 | 2.82 | 96 | 4 |

The surface area of the metal-doped catalysts was in the same range as that of pure Aluminum-silicate catalysts. This observation suggests the enhanced activity may be due to a synergistic effect between the basic matrix and the metal dopant.

Moreover, Ni doping on basic Al/Si shows the highest activity of all synthesized materials. This unusual activity might be related not only to Ni presence but also to a superior surface area and the presence of stronger basic sites measured for this complex, as shown in the characterization section. These findings align with the literature, highlighting the exceptional efficacy of zinc [21] and nickel-loaded [23,24] matrices in catalytic cracking for hydrocarbon fuel production. Nickel, in particular, is renowned for its remarkable carbon adsorption capabilities and faster carbon diffusion rates, making it a highly desirable catalyst component, with additional benefits such as lower cost and toxicity.

Finally, the presence of metals in addition to optimizing the reaction temperature and using a catalyst, improves the selectivity towards liquid products, leading to even higher yields and potentially reducing the formation of unwanted products.

## 4. Conclusions

This study explored the potential of metal-doped basic and acidic Al/Si matrices as catalysts for the catalytic cracking of waste motor oil. Specifically, the effectiveness of Mg,

Zn, Cu, and Ni as metal dopants was investigated in these matrices. The study's results revealed that the metal-doped aluminum silicates exhibited significantly higher conversion rates during catalytic cracking compared to superficial thermal cracking and acidic or basic aluminum silicates. Among the metal dopants tested, Ni impregnation under basic Al/Si matrices demonstrated the best catalytic performance, with a three-fold increase in conversion yield compared to experiments without metal doping. This material showed the highest specific surface area and the presence of stronger basic sites compared to the pristine Al/Si, due to the alkali treatment and metal doping.

Interestingly, the selectivity to products was not significantly affected by the catalytic and/or metal doping. This suggests that the metal dopant used in the catalyst has minimal impact on the final products obtained during the catalytic process.

Furthermore, the final products obtained from all the catalytic processes were in the form of a liquid product that meets the requirements set by the ASTM characterization methods for a diesel-like fuel. This finding highlights the potential of using waste motor oil as a feedstock to produce high-quality fuels through catalytic cracking. Overall, the study provides valuable insights into developing efficient and sustainable catalytic processes for the valorization of waste materials.

**Supplementary Materials:** The following supporting information can be downloaded at: https://www.mdpi.com/article/10.3390/su151310522/s1, Figure S1: Scanning electron microscopy with energy-dispersive X-ray spectroscopy (SEM-EDX) analysis (+) of a Cu-doped aluminosilicate, Figure S2: $N_2$ adsorption–desorption isotherms of synthesized materials., Figure S3: Thermal gravimetric analyses of synthesized materials, Figure S4: Infrared spectra of synthesized materials, Table S1: Mass balance of thermal and acidic/basic catalytic cracking process, Table S2: Mass balance of acidic/basic catalytic cracking process doped with different metals.

**Author Contributions:** D.A.S. and A.H. were responsible for the study conception and design. Material preparation, data collection and analysis were performed by A.A. and D.A.S. Characterization of the catalysts was performed by A.G.-C. The first draft of the manuscript was written by S.P. All authors have read and agreed to the published version of the manuscript.

**Funding:** This work was supported by Universidad San Francisco de Quito POLI Grants 2016 Nr. 11220 and Collaboration Grants 2023.

**Institutional Review Board Statement:** Not applicable.

**Informed Consent Statement:** Not applicable.

**Data Availability Statement:** Not applicable.

**Acknowledgments:** A.G.-C. acknowledge the support given by the research group of Materials for Medicine and Biotechnology of the Institute of Materials Science of Madrid.

**Conflicts of Interest:** The authors declare no conflict of interest.

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
