# Peer review of "Chemical Recycling of Used Motor Oil by Catalytic Cracking with Metal-Doped Aluminum Silicate Catalysts"

_sustainability, doi:10.3390/su151310522_

Round 1
Reviewer 1 Report
-The introduction is too short and does not adequately address the state of the art. The initial part may be acceptable, as it reviews the issue of contamination caused by used oil and mentions various alternatives. However, when it comes to discussing catalytic cracking to obtain diesel and gas, only one reference (17) is mentioned. While it is possible that catalytic cracking has not been specifically used for valorizing used oil, it has been widely employed for similar organic hydrocarbons and other waste materials. Therefore, a more extensive introduction that contextualizes the work should be rewritten.
-Why are you using those metals (Mg, Zn, Cu, and Ni)? Citations about those metal in similar processes should be included also.
-You should use always °C as Celsius symbol since you are using in some cases ºC or oC
-After analyzing everything related to the surface area, it took me quite some time to grasp the results. In the case of the isotherms presented in the supplementary material, I would divide the data into two graphs: one for the base series and another for the acid series. Additionally, I would also include the isotherms of the nickel catalysts there.
-Line 184-186 “Interestingly, most metal-doped materials changed the catalyst properties by reducing their surface area and pore diameters compared to Al/Si catalysts.” This general statement, but 2 of the catalyst have a higher surface area (B-Al/Si-Mg and B-Al/Si-Ni). Moreover, two other catalysts (A-Al/Si-Mg and A-Al/Si-Ni) have a higher pore volume than B-Al/Si.
-Furthermore, the effect of the incorporated metal and the acid or base procedure on the surface area and porosity are worth investigating. It is intriguing how materials synthesized in different ways (A-Al/Si and B-Al/Si-Mg) exhibit such similarity in surface area pore volume and adsorption-desorption hysteresis. Moreover, it appears that there is no apparent relationship between the synthesis conditions (acid or base), and the metal with the properties of the catalysts.
- The characterization performed is not sufficient to fully characterize the material and discuss the catalytic results. A more comprehensive characterization of the catalysts could provide valuable insights into the catalytic behavior. The total acidity, acidity strength, and type of acidity of the materials have not been analyzed. Considering that acidity/basicity is a crucial factor, a more detailed examination of these properties could provide a better explanation for the results obtained with these catalysts. Furthermore, it would be beneficial to investigate how the metals affect the acidity of these materials. Techniques such as XPS could provide a better understanding of the nature of the doped metal, or other methods that could shed light on the discussion of the results.
- In regard to the results and discussion, the results are presented but not adequately discussed. The obtained results are simply described without a clear explanation of why they occur (lines 277-294). It is unclear why the incorporation of metals in B-AlSi significantly improves activity, while the incorporation of metals in A-AlSi does not result in notable improvement except for A-AlSi-Zn. Additional characterization that provides a better understanding of how the metal is incorporated, possible incorporation into the aluminosilicate structure, catalyst acidity, oxidation state, etc., could be of great help.
-Moreover, incorporating relevant references that could help in the analysis of the results of catalytic cracking processes under similar conditions would also contribute to the discussion.
Author Response
Please find attached our point-to-point responses.

Reviewer 2 Report
1) It is recommended that discuss more information and the catalytic importance of aluminum silicates catalysts in the introduction section.
2) Should provide the particle size of the Cu-doped acid catalyst also.
3) Should provide the explanation, why the morphology is different from Cu-doped basic catalyst and Cu-doped acid catalyst.
4) According to reported data, metal-doped materials are reduced in their surface area and pore diameters compared to aluminum silicates catalysts. But, among all catalysts, Ni-doped aluminum silicates showed the best catalytic performance, with conversions and yields three times higher than metal-free catalysts. Explain?
Rephrase references 15 and 16 as per the journal format
In the current manuscript, the authors have described the chemical recycling of used motor oil for catalytic cracking with aluminum silicate catalysts. The authors have shown the different characterization techniques for synthesized materials and the cracking process. Using the waste motor oil as a feedstock for the production of high-quality fuels through catalytic cracking. Therefore, this reported protocol is useful to develop for the sustainable catalytic processes, hence, after minor modifications, the protocol could be interesting for readers and may be published in the journal.
Author Response

(The authors gave the same response as above.)

Reviewer 3 Report
The manuscript (sustainability-2368002) reported an interesting metal doped aluminum silicate for the cracking of used motor oil and found the beneficial effect of Ni doping. The topic is interesting and has great significance. Before the consideration for acceptance, the following comments should be addressed.
1. Please offer the description of reaction mechanism for the waste oil cracking using aluminum silicate and tell the reader what is the active sites in the catalyst in the introduction part.
2. The reason why the author try to improve the catalytic performance by metal doping is not included in the introduction.
3. Please analyze why the specific surface area differs significantly for the doped samples?
4. There are many grammar mistakes such as "The morphology of the synthesized catalysts was analyzed via SEM imaging. See 171 exemplary images of Cu-doped basic and acid catalysts in Figure 2." Please check the whole manuscript and make corresponding corrections.
4. What is the catalyst used in the catalytic results of Figure 5.
5. Please offer the evidence of the metal doping in the framework of aluminum silicate rather than forming the seperated metal catalyst, e.g. Ni or NiO. XRD characterization is recommended to be carried out.
6. Why Ni doping is most active compared to other catalyst? Is Ni site active for this reaction?
7. Please pay attention to the statement of "with conversions and yields three times higher than metal-free catalysts." since there is metal contained in the reference catalyst.
8. Metal doping could play important role for the introduction of defective sites and help boosting the intrinsic catalytic activity. The following references are recommented for the citation: Journal of Hazardous Materials, 452 (2023) 131319, Applied Surface Science, 2022, 600, 15404
The quality of Engnish Language could be further improved.
Author Response

(The authors gave the same response as above.)

Round 2
Reviewer 3 Report
The comments have been well addressed, the manuscript is suggested for the acceptance.
Author Response
Thanks for the possitive comment about our work.